# Early Administration of Intravenous Hydration and Opioid Analgesics Is Correlated with Decreased Admission Rates during Vaso-Occlusive Episodes in Sickle Cell Disease

**DOI:** 10.3390/jcm13071858

**Published:** 2024-03-23

**Authors:** Bowon Joung, Ethan Miles, Farris Al-Manaseer, Won Jin Jeon, Darren Wijaya, Jin Hyun Moon, Philip Han, Jae Lee, Akhil Mehta, Alan Tseng, Kaylin Ngo, Huynh Cao, Esther G. Chong

**Affiliations:** 1Department of Internal Medicine, Loma Linda University Medical Center, Loma Linda, CA 92350, USA; bjoung@llu.edu (B.J.); emiles@llu.edu (E.M.); wjjeon@llu.edu (W.J.J.); dwijaya@llu.edu (D.W.); jmoon1@llu.edu (J.H.M.); 2School of Medicine, Loma Linda University, Loma Linda, CA 92350, USAjlee36@students.llu.edu (J.L.); 3Division of Hematology and Oncology, Houston Methodist Hospital, Houston, TX 77030, USA; 4Division of Hematology and Oncology, Loma Linda University Medical Center, Loma Linda, CA 92354, USA; atseng@llu.edu (A.T.);; 5Department of Biological Science, University of California San Diego, La Jolla, CA 92093, USA; k5ngo@ucsd.edu

**Keywords:** sickle cell disease, vaso-occlusive episodes, IV pain management, IV hydration, IV opioid, clinical outcome, hospital admission

## Abstract

**Background:** Painful vaso-occlusive episodes (VOEs) are the hallmark of sickle cell disease (SCD) and account for frequent visits to the emergency department (ED) or urgent care (UC). Currently, the early administration of analgesics is recommended as initial management; however, there is a need for further understanding of the effect of prompt analgesics and hydration during VOEs. The objective of this study is to analyze the factors associated with the rate of hospital admission in the setting of time to intravenous (IV) analgesics and hydration. **Method:** This retrospective single-institution study reviewed adult and pediatric patients with SCD who presented with VOEs from January 2018 to August 2023. **Results:** Of 303 patient encounters, the rates of admission for the overall group, the subgroup which received IV hydration within 60 min of arrival, and the subgroup which received both IV analgesics and hydration within 60 min were 51.8%, 25.6% (RR = 0.46), and 18.2% (RR = 0.33), respectively. Further, factors such as gender and the use of hydroxyurea were found to be significantly associated with the rate of admission. **Conclusions:** This signifies the importance of standardizing the management of VOEs through the timely administration of IV analgesics and hydration in both adult and pediatric ED/UC.

## 1. Introduction

Sickle cell disease (SCD) is a multi-system hematologic disorder, manifested by extremely painful and recurrent vaso-occlusive episodes (VOEs) [1]. VOEs result from polymerization hemoglobin S (HbS) induced by a multitude of triggers including infection, fever, dehydration, anemia, and even socio-economic stressors [2]. This pathologic polymerization causes red blood cell (RBC) rigidity and increased adhesion to vessels, resulting in endothelial dysfunction, microvascular occlusions, and impeded blood flow [2]. Although most patients with SCD initially present with uncomplicated VOEs where administration of oral or intravenous (IV) analgesics resolves their pain, other serious medical conditions may lead to hospital admissions if these recurrent VOEs are not well managed [3]. For example, many components of complicated VOEs, which include acute chest syndrome, splenic infarction, stroke, and severe anemia, are acute onset and life-threatening, and contribute to high morbidity and mortality in patients with SCD [3].

In addition to a lower quality of life and significant morbidity, the numerous hospital visits from VOEs generate tremendous financial burden. In 2023, the annual total expenditure for SCD-related emergency department (ED) visits and hospitalizations in the United States was estimated to be more than one billion dollars [4]. Given that almost ninety percent of SCD hospital admissions are primarily due to VOEs in acute ED or urgent care (UC) settings, the treatment of SCD has focused on rapid symptom management and prevention of future episodes [5]. In outpatient settings, various disease-modifying therapies, such as hydroxyurea, L-glutamine, crizanlizumab, and voxelotor, have shown their effectiveness in reducing the frequency of VOEs [5,6,7]. However, many patients with SCD demonstrate suboptimal adherence to these medications due to multi-level barriers related to the complex health care system, resource scarcity, and psycho-social issues. This overall poor adherence to disease-modifying medications results in the inability to curb the frequency of VOEs [6].

Current guidelines from The National Heart, Lung, and Blood Institute (NHLBI) and American Society of Hematology (ASH) recommend the time-sensitive administration of analgesics including a tailored dosage of opioid analgesics within one hour of arrival to the ED [8,9]. Despite the presence of nationwide evidence-based guidelines, there is not enough emphasis on the use of standardized analgesics or hydration protocol to manage VOEs in acute care settings. Considering this, our study sought to understand the significance of the prompt administration of IV analgesics and IV hydration during VOEs and its impact on clinical outcomes, specifically hospital admissions.

## 2. Materials and Methods

This institutional review board-approved retrospective observational cohort study was conducted at a single academic tertiary center involving both adult and pediatric EDs and UC. All patients who presented to this center for acute sickle cell VOEs were screened for inclusion from January 2018 to August 2023 (Table 1). 

Patients were initially selected according to the International Classification of Diseases, Ninth Revision, Clinical Modification (ICD-9-CM) and Tenth Revision (ICD-10-CM) codes for SCD with acute crisis (ICD-9-CM 282, ICD-10-CM D57, D57.4, D57.219). Only patients who received both IV analgesics and IV fluids were included. Each encounter was reviewed to obtain demographics including age, gender, race, and previous use of hydroxyurea. ED/UC management of VOEs was analyzed including time from initial presentation to the first dose of IV analgesics and IV hydration. The primary endpoint was the rate of admission, which was calculated as the number of hospital admissions divided by the number of acute care visits, either to the ED or UC. 

Furthermore, the impact of gender on admission and length of hospital stay was investigated. This involved calculating admission proportions for each gender using the chi-squared test. Then, an independent samples *t*-test was conducted to compare the mean length of stay between male and female patients. In addition to gender analysis, the influence of hydroxyurea on hospital admission and length was also examined. A chi-squared test was used to assess the association between hydroxyurea use and admission status. Using a contingency table, the effect of hydroxyurea use on length of hospital stay was then investigated with an independent sample *t*-test and effect size measures were calculated to determine practical significance of observed differences.

Statistical evaluation was completed using the python SciPy library. Categorical variables were evaluated with the chi-squared test. Relative risk (RR) and 95% confidence intervals (CIs) were calculated for the admission rate. Continuous variables were compared using Welch’s T-test and the Mann–Whitney U-test based on the normality of the variable. Statistical significance was set at *p*-value < 0.05. 

## 3. Results

A total of 303 patient encounters met the inclusion criteria and were included in the study. These encounters involved 93 patients, consisting of 51 adults and 42 pediatric patients. The median age of all the patients was 20 years. For adults and pediatrics, the median age was 28 and 12, respectively. The demographic characteristics of the admitted and non-admitted patients are shown (Table 2). Out of the total 303 encounters, 252 encounters were managed with IV opiates whereas 51 encounters were managed with IV non-opiates. The overall admission rate among the total encounters was 51.8%. 

The admission rates based on the timing of IV hydration and IV analgesics are shown in Figure 1. The admission rate was significantly lower in the patients who received IV analgesics within the first 60 min [30.2%, RR 0.54, 95% CI 0.35–0.82], IV hydration within the first 60 min [25.6%, RR 0.46, 95% CI 0.27–0.80], and both IV analgesics and hydration within the first 60 min [18.2%, RR 0.33, 95% CI 0.52–0.68] (Figure 1a).

In adult patients, the admission rate of those who received IV pain medication within 60 min upon ED encounter was 37.0% compared to 61.7% for those who did not receive IV pain medication within 60 min (*p*-value = 0.019). The admission rate of adult patients who received IV hydration within 60 min upon ED encounter was 23.5%, while it was 61.4% for those who did not receive IV hydration within 60 min (*p*-value = 0.003). In pediatric patients, the admission rate of those who received IV pain medication within 60 min of ED encounter was significantly lower, at 29.2%, compared to 55.5% for those who did not receive IV pain medication within 60 min upon ED encounter (*p*-value = 0.018). The admission rate of pediatric patients who received IV hydration within 60 min upon ED encounter was 31.6% and those who did not receive IV hydration within 60 min was 54.0% (*p*-value = 0.068) (Figure 1b). 

In the admission arm, 10.3% of the patient encounters received IV interventions within 60 min (17 out of 165). In the non-admission arm, 24.6% of the patient encounters received IV interventions within 60 min (34 out of 138) (Figure 1c). The patients who were not admitted received IV hydration and IV opiates sooner than those who were admitted by 88.1 min and 39.2 min, respectively (Table 3). There was a significant correlation between the time to first IV hydration and IV analgesics (r(311) = 0.749, *p*-value < 0.001). 

There was a significant association between hydroxyurea use and hospital admission. The chi-squared statistic was found to be at 12.7, with a corresponding *p*-value of 0.0004 (α = 0.05). The independent samples *t*-test for length of admission yielded a t-statistic of 2.6 and a *p* value of 0.0095, indicating a statistically significant difference in the length of admission between the two groups. When evaluating gender and admission, the chi-squared statistic obtained was 10.66 with a *p*-value of 0.001. This indicates a significant association between gender and admission. 

The mean number of days of hospital admission was 5.6 and 6 days for each adult and pediatric patient (*p*-value = 0.63). Upon further analysis of the length of hospital stay and gender, the t-statistic was found to be 0.48 with a *p*-value of 0.63. There was no significant difference in the length of stay between the genders. 

Lastly, a total of nine patients were identified whose encounters to ED were dated within 2 weeks apart from each other. The admission rate of the encounters made by these patients who had frequent ED visits (N = 100) was 67% and the admission rate of the encounters made by those except the patients with the frequent ED visits (N = 203) was 48.3% (*p*-value = 0.002). 

## 4. Discussion

The optimization and standardization of clinical practice for the management of VOEs have the potential to bring about positive clinical outcomes, including the prevention of unnecessary hospital admissions and an improvement in the overall quality of life in patients with SCD. In the approach to treating acute pain during VOEs, it is important to understand that VOEs are a multifactorial disease process, including not only sickling of RBCs but also inflammation, endothelial dysfunction, adhesion, and multicellular aggregation [6,10]. The result of these pathological pathways is compromised blood flow to individual organs throughout the body, causing acute and chronic pain of various types [6,7,8,10]. The goal of VOE treatment is to halt and delay the progression of these pathologic steps and to rapidly curb pain [10]. 

Many studies have examined improvements in the current structural and clinical barriers against the management of VOEs. The institutionalization of a healthcare center dedicated to providing personalized care for patients with SCD is one of these recently developed methods. A sickle cell infusion clinic (SCIC), affiliated with the Johns Hopkins Hospital, showed a significant reduction in hospital admission rates from the ED when compared to the year prior to institutionalization and the year after [11,12,13]. The ESCAPED trial is an ongoing multi-center prospective cohort study, which has so far shown a substantial reduction in hospitalizations for those who were treated in SCICs compared to those who were treated in the ED (relative risk reduction (RRR) of 75%, *p* < 0.001) [14]. However, this healthcare system design is not always plausible in certain communities and demographics. Our single-institution retrospective study characterizes the clinic outcomes of sickle cell patients based on their initial management in ED/UC settings, where a standardized protocol to manage VOEs was not yet implemented. 

Our study confirmed that the early use of IV analgesics during VOEs in acute care settings is linked to a statistically significant improvement in clinical outcomes, one of which is the prevention of hospital admissions. Our data also showed that patients who received IV hydration within 60 min in an acute care setting displayed a significant reduction in hospital admission rates compared to those who did not receive early IV hydration. In acute care settings, IV fluids were frequently given during VOEs with the expectation of slowing down RBC sickling and augmenting pain relief [8]. 

However, it is difficult to discern the individual effect of IV hydration and IV analgesics since the time to IV hydration and IV analgesics were strongly correlated. Current guidelines have maintained a neutral viewpoint in using IV hydration. The rationale for the use of IV hydration is that intracellular dehydration is known to trigger and prolong the sickling of RBCs. Several studies have observed that the initial glycation of hemoglobin leads to the dehydration of red blood cells [10]. Few case reports have suggested that the use of rapid administration of hypotonic IV fluid during VOEs is superior in preventing SCD-related renal dysfunction and reducing the frequency of VOEs compared to the use of isotonic IV fluid [15,16,17]. This study also found an additive effect on the reduction in hospital admission rates when IV analgesics and IV hydration were both administered within 60 min compared to when either was individually given early. 

On the other hand, other studies have reported several adverse effects in the overuse of IV fluid resuscitation for managing VOEs. In particular, with unregulated IV fluid administration, clinicians should be cautious of numerous adverse effects, mostly manifested as hypervolemic states, such as pulmonary edema, dyspnea, hypoxia, atelectasis, and even acute chest syndrome [18,19]. Unfortunately, our study did not further investigate the volume and rate of fluid infusion in each patient encounter due to the complexity of the clinical presentations of patients with a variation in age, weight, height, and initial volume status. This paper further acknowledges that there are not enough clinical data to fully support the efficacy of IV fluid in the resolution of cellular dehydration during VOEs. Even though the efficacy of IV fluids is still controversial, this paper emphasizes the need for standardized tools to judiciously administer IV fluids in conjunction with IV analgesics during the initial stage of clinical presentation in the acute care setting.

Regarding the significance of differing demographics such as age, in both pediatrics and adult patients, the earlier administration of IV analgesics has resulted in a significant reduction in hospital admission. However, the earlier administration of IV hydration resulted in fewer admissions only in adult patients, not in pediatric patients. Also, the admission rate from patients who had frequent ED visits was significantly higher than that of the patients who did not have frequent ED visits (67.0% vs. 48.3%, respectively, *p*-value = 0.002). These findings suggest that the progression and severity of SCD also play a large role in whether a patient is more likely to be admitted or not. The observed significant association between hydroxyurea use and admission highlights the potential impact of medication on disease management and healthcare utilization. This may improve clinical outcomes, leading to reduced admission and associated healthcare costs. Similarly, the identified association between gender and admission emphasizes the importance for gender-specific factors and disparities in healthcare delivery. 

It was interesting to observe that in our study, the median time for administration of IV hydration and analgesics in the admission arm exceeded 60 min upon arrival to the ED. The time difference between the admission arm and the non-admission arm for each of IV hydration and IV analgesics was significant only by about 40 min and 90 min, respectively. Also, large deviations were noted among the times recorded for these IV interventions. These findings suggest that clinicians exhibit a variation in their practices in managing VOEs and there may be an institutional and systemic barrier to following current guidelines for prompt IV intervention. Thus, we propose further research in multi-institutional settings to characterize and validate the relationship between prompt IV intervention and clinical outcomes during VOEs. 

The limitations of this study include its retrospective nature and the heterogenous management of patients, reflected in the wide range in the time to first IV analgesic and hydration and the lack of standardized dosing and type of IV analgesic. Additionally, this study used admission as its single clinical endpoint and did not evaluate other clinical outcomes, including the readmission rate. Future prospective studies should standardize analgesics and IV fluid type and rate. Further studies can provide insights into the relationship between hydroxyurea use, gender, and hospital admission. Investigating the impact on disease progression and quality of life can provide an understanding of treatment and prognosis. 

## 5. Conclusions

Our study demonstrates that the early administration of IV analgesics and hydration in patients with SCD during VOEs is linked to a significant reduction in hospital admission rates. Although randomized trials or prospective studies are required to corroborate our results, our findings support the importance of providing standardized care per current guidelines and prioritizing rapid interventions, specifically IV analgesics and IV hydration, during VOEs to improve their clinical outcomes.

## Figures and Tables

**Figure 1 jcm-13-01858-f001:**
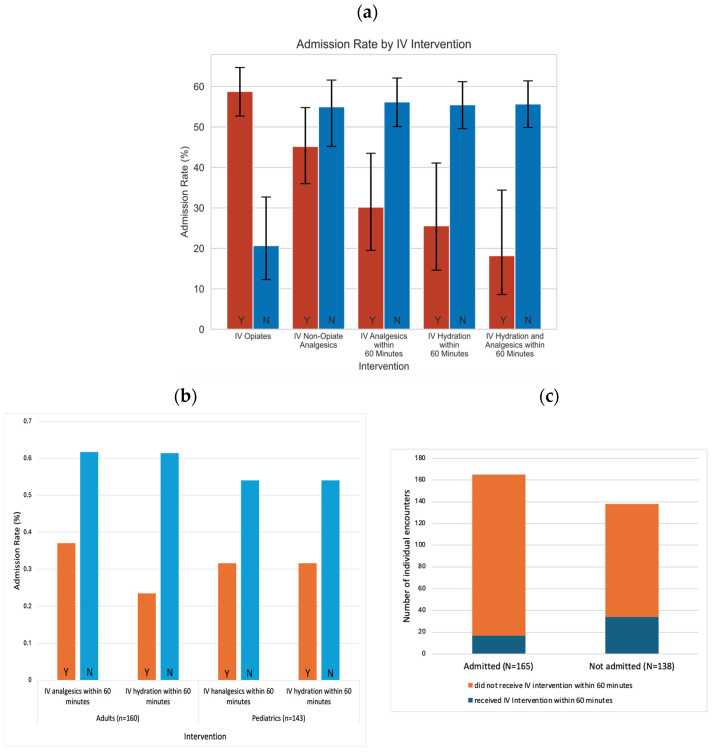
(**a**) Admission rate by IV analgesics and IV hydration within 60 min in all patient encounters. (**b**) Admission rate by IV analgesics and IV hydration in adults and pediatrics patients. (**c**) Number of patient encounters by hospital admissions and the proportions of encounters with IV intervention within 60 min.

**Table 1 jcm-13-01858-t001:** Inclusion and exclusion criteria.

Inclusion Criteria	Exclusion Criteria
Diagnosis of SCD including both genotype SS and S-Beta thalassemia. Chief complaint or reason for admission documented as either “acute sickle cell crisis” or “sickle cell pain crisis” under ICD-9/10-CM codes. Patients who had a history of multiple ED visits within 2 weeks of each other were accounted for.	Sickle cell trait or other genotypes besides SS and S-Beta thalassemia. Chief complaint or reason for admission including acute chest syndrome, stroke, or other secondary diagnoses (complicated VODs). Patient not receiving both IV hydration and pain medication. Patient encounters involving only oral (PO) pain medication, not in IV form, were excluded. Patients leaving against medical advice.

**Table 2 jcm-13-01858-t002:** Demographics of patient encounters.

	Admitted (n = 165)	Not Admitted (n = 138)	*p*-Value
Gender (%)	Male 88 (53.3%) Female 77 (46.7%)	Male 42 (30.4%) Female 96 (69.6%)	<0.001
Age (%)	Adults 92 (55.8%) Pediatrics 73 (44.2%)	Adults 68 (49.3%) Pediatrics 70 (50.7%)	0.26
Race (%)	Black 159 (96.4%)	Black 133 (96.4%)	0.23
	Hispanic 6 (3.6%)	Hispanic 3 (2.2%)	
	Other 0 (0%)	Other 2 (1.4%)	
Hydroxyurea (%)	Yes 97 (58.8%)	Yes 52 (37.7%)	<0.001
	No 68 (41.2%)	No 86 (62.3%)	

**Table 3 jcm-13-01858-t003:** Average time to receive first IV analgesics or IV hydration in patient encounters that were admitted and that were not admitted.

	Not Admitted	Admitted	Difference
IV analgesics In minutes (SD)	145.3 min (142.2)	184.5 min (154.5)	39.2 min (*p* = 0.002)
IV hydration In minutes (SD)	153.3 min (149.5)	241.1 min (213.8)	88.1 min (*p* < 0.001)

SD = standard deviation.

## Data Availability

Data are contained within the article. The raw data supporting the conclusions of this article will be made available by the authors on request.

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
