# Peer review of "Early Administration of Intravenous Hydration and Opioid Analgesics Is Correlated with Decreased Admission Rates during Vaso-Occlusive Episodes in Sickle Cell Disease"

_jcm, 2024, doi:10.3390/jcm13071858_

Round 1

Reviewer 1 Report

Comments and Suggestions for Authors

Dear Authors,

Review its own pratice is the better way to improve the management of our patient (not only yours but all sickle cell patients) and I would like to congratulate you for this work.

You will find my comments below.

Regarding the demographic data, is it possible to add the median age of the all population and separetely of adults and pediatrics patients ? and also the difference between adult and pediatric cohort in the table 2 ?

Regarding your population, based on Table 2, it appears that patients admitted and not admitted differ from gender and also from chronic treatment with HU. Can you comment on that in the discussion part ? Do you think that as the not admitted patients are less treated with HU, it could be possible that this is a « less severe » population and it could interfer with the decision to admit or not the patient ? Did you made multivariate analysis on this purpose ?

As wrote in the results part, IV hydratation seems to be 500 to 1000ml for adults and 500ml for pediatric patients. However, can you specifiy in how many times the IV hydratation is given ? Indeed, 500ml for pediatric patient with 15-20kg seems to be a lot of and could increase the risk of overload (as discussed). As it is a retrospective study, I understand that is not possible to be more precise in the methodology part. But once again, if we have the age (and the median wheight) of the patient and also the median time of administration (1 hour, 2 hour or more), it will be more easy to assess the risk of overload.

Did you find the same difference about admission rate and the time of IV medication and IV hydratation for adults and for children ? If not, can you comment on that ?

313 patient encounters in 93 patients in 5 years is not a lot (less than 1 per year per patient). How many sickle cell patients are followed in your center ? How many patients (or pain episodes) have been excluded from this analysis (due to only oral medication or lack of data) ? Is it possible to have a « flow chart » of your selected population ?

Do you have data about the treatment received at home (before ED/UC visit) ?

For admitted patients, do you have data about the duration of hospitalisation ?

If possible, it could be greatful to know if the patient admitted are the same who were not admitted 2 days before. Because the interpretation could be very different if the majority of admitted patient could go back home a couple of day before.

Finally, in Table 3, you show the time to receive first IV analgesics or hydratation. I suppose it is the median time from ED presentation. However for both this time is around 2.5 hours for not admitted patients and more than 3 hours for admitted patients (much more than 60 min expected). Can you comment on that ? How many patient have been in each arm (admitted with IV less than 60 min, admitted with more than 60 min, not admitted with IV less that 60 min, not admitted with more thant 60 min) ?  Did you already have the opportunity to organize a dedicated way to take in charge sickle cell patient in your center ?

Author Response

Dear. Reviewer

I appreciate for your insights and feedback. I apologize for the delay in responses. we had to do the major change with data collection as well as statistical analysis. 

Regarding the demographic data, is it possible to add the median age of the all population and separetely of adults and pediatrics patients ? and also the difference between adult and pediatric cohort in the table 2 ?

We added the new median ages for the overall patient populations as well as for adults and pediatrics. We update Table 2. 

Regarding your population, based on Table 2, it appears that patients admitted and not admitted differ from gender and also from chronic treatment with HU. Can you comment on that in the discussion part ? Do you think that as the not admitted patients are less treated with HU, it could be possible that this is a « less severe » population and it could interfer with the decision to admit or not the patient ? Did you made multivariate analysis on this purpose ?

In order to address the question, we added sub group analysis for admission based on previous use for hydroxyurea or age. Please refer to line 149-155 (result) and line 218-229 (discussion) 

As wrote in the results part, IV hydratation seems to be 500 to 1000ml for adults and 500ml for pediatric patients. However, can you specifiy in how many times the IV hydratation is given ? Indeed, 500ml for pediatric patient with 15-20kg seems to be a lot of and could increase the risk of overload (as discussed). As it is a retrospective study, I understand that is not possible to be more precise in the methodology part. But once again, if we have the age (and the median wheight) of the patient and also the median time of administration (1 hour, 2 hour or more), it will be more easy to assess the risk of overload.

Unfortunately, after another round of data mining, there was a wide range of weights, age, and height differences among even the pediatrics and adults, where approximating the volume/ rate of IV fluid given at ED was not plausible. 

Did you find the same difference about admission rate and the time of IV medication and IV hydratation for adults and for children ? If not, can you comment on that ?

We did add a new statistical analysis for admission rate for adults and pediatrics. please refer to line 110 to 119 and figure 1b.

313 patient encounters in 93 patients in 5 years is not a lot (less than 1 per year per patient). How many sickle cell patients are followed in your center ? How many patients (or pain episodes) have been excluded from this analysis (due to only oral medication or lack of data) ? Is it possible to have a « flow chart » of your selected population ?

Initially we had about 540 patients who were admitted to ED for acute pain crises in the past 5 years. However, with the inclusion criteria, the number of patient encounters has been finalized to 303. 

Do you have data about the treatment received at home (before ED/UC visit) ?

Unfortunately further data about what type of pain medication was tried prior to ED was not available in the medical document. 

For admitted patients, do you have data about the duration of hospitalisation ?

Yes, we added this information to our revised draft. please refer to line 156-159

If possible, it could be greatful to know if the patient admitted are the same who were not admitted 2 days before. Because the interpretation could be very different if the majority of admitted patient could go back home a couple of day before.

Yes, thank you. We added the definition of "Frequent Flyers" as Patients who had a history of multiple ED visits within 2 weeks of each other were accounted for. We intentionally avoided using the term frequent flyers because in the literature it has been noted that it holds the derogatory terminology. 

Finally, in Table 3, you show the time to receive first IV analgesics or hydratation. I suppose it is the median time from ED presentation. However for both this time is around 2.5 hours for not admitted patients and more than 3 hours for admitted patients (much more than 60 min expected). Can you comment on that ? How many patient have been in each arm (admitted with IV less than 60 min, admitted with more than 60 min, not admitted with IV less that 60 min, not admitted with more thant 60 min) ?  Did you already have the opportunity to organize a dedicated way to take in charge sickle cell patient in your center ?

Yes, I added the distribution of IV intervention either < 60 min or > 60 min for admitted and not admitted arms is now updated on Figure 1c. Line 120-122.

Reviewer 2 Report

Comments and Suggestions for Authors

This study includes a substantial number of subjects, but as presented, it is rather limited and makes a rather modest addition to the literature.

Specific comments:

1. Among the 313 patient encounters, there were only 93 distinct individuals. The authors need to reanalyze the data breaking out the “frequent flyers”. Do they have a higher or lower rate of admission than one time subjects?

2. Where is the information on ages?

3. Is there a statistical comparison regarding sex or hydroxyurea use? (They both look different to me.) they might suggest additional explanations.

4. Are there additional clinical and laboratory data for the subjects? Especially things like initial pain scores, home pain meds, etc.

5. Any assessment of initial hydration status?

6. The authors should discuss controversies or lack of data regarding IV fluid management in SCD.

Author Response

Dear Reviewer 2, 

I truly appreciate your insights and feedback. 

We tried to reflect your comments and thus completed another round of data mining and statistical analysis. 

  1. Among the 313 patient encounters, there were only 93 distinct individuals. The authors need to reanalyze the data breaking out the “frequent flyers”. Do they have a higher or lower rate of admission than one time subjects?

We updated our data and reported a number of “frequent flyers.” We avoided using the term frequent flyers because the literature has mentioned that it holds the derogatory terminology against sickle cell patients. Please refer to line 160-164 (result) and  line 221 to 224 (Discussion), there was statistically higher admission rate amongst the encounters resulted from patients who had frequent ED visits. 

  1. Where is the information on ages?

The median age of all patients was 20 years. For adults and pediatrics, the median age was 28 and 12 respectively. The demographic characteristics of admitted and non-admitted patients are updated in Table 2.

  1. Is there a statistical comparison regarding sex or hydroxyurea use? (They both look different to me.) they might suggest additional explanations.

We added subgroup analysis of admission on sex or hydroxyurea use. Please refer to line 149 to 154 (result) as well as line 218 to 230 (discussion) 

  1. Are there additional clinical and laboratory data for the subjects? Especially things like initial pain scores, home pain meds, etc.

Unfortunately, we could not obtain detailed pain scores documented in patient charts. Hydroxyurea is considered as one of the home medications that patients take and we hope to use it as a possible associating factor for admission. 

  1. Any assessment of initial hydration status?

Clinical sign and symptoms for initial hydration was difficult to assess and clearly define based on the documentation. 

  1. The authors should discuss controversies or lack of data regarding IV fluid management in SCD.

I agree with your point. Please refer to line 206 to 217 for added insights regarding current literature regarding IV fluid use for acute pain crises. 

Round 2

Reviewer 2 Report

Comments and Suggestions for Authors

Can the authors add any insight (or even speculation) regarding why some patients got IV hydration or pain meds faster than others?

Is this just chance or business of the ED? Did some patients initially seem less sick but then get worse and eventually need admission?

Are any of the repeated patients represented in both the discharge and admit groups? Can they be compared?

Author Response

Dear Review 2,

Thank you again for your prompt response and comments. Below are our responses to your comments. 

1. ​​Can the authors add any insight (or even speculation) regarding why some patients got IV hydration or pain meds faster than others? Is this just chance or business of the ED? Did some patients initially seem less sick but then get worse and eventually need admission?

One of the speculations is that due to the lack of standardization and guideline-based management of vaso-occlusive episodes related to SCD, differing emergency room or urgent care physicians and providers have varying practice styles. Our study highlights the need for standardized, fast-tracked regimens, similar to the management of other acute disorders such as myocardial infarction and sepsis. 

In terms of admission, given that our study excluded those with complicated VOEs, such as acute chest syndrome, our speculation is that patients were admitted for other indications such as intractable pain. Thus, the admission criteria may vary by each clinician and also stress the importance of implementing the standardized care for sickle cell patients who undergo VOEs in any acute care center (ED or UC). However, this could be very well another area that we may evaluate in the future.

2. Are any of the repeated patients represented in both the discharge and admit groups? Can they be compared?

There are repeated patients as our study evaluates patient encounters rather than individual patients. Though we have not compared the repeated patients, each patient encounter, even for the same patient, should be considered as individual encounters. The focus of our study is more on the ED/ UC management of VOEs. Having said that, future directions can include further analysis on the characteristics of patients who repeatedly present to the ED/ UC or are frequently admitted.

Please let us know if there are any other parts that should be further addressed. We truly appreciate your time. 

Attached is the new version of manuscript with the updated authorship. 

Sincerely

Bowon Joung